# Phonophoresis through Nonsteroidal Anti-Inflammatory Drugs for Knee Osteoarthritis Treatment: Systematic Review and Meta-Analysis

**DOI:** 10.3390/biomedicines10123254

**Published:** 2022-12-14

**Authors:** Francisco Javier Martin-Vega, David Lucena-Anton, Alejandro Galán-Mercant, Veronica Perez-Cabezas, Carlos Luque-Moreno, Maria Jesus Vinolo-Gil, Gloria Gonzalez-Medina

**Affiliations:** 1Department of Nursing and Physiotherapy, Faculty of Nursing and Physiotherapy, University of Cadiz, 11009 Cadiz, Spain; 2Department of Physiotherapy, Faculty of Nursing, Physiotherapy and Podiatry, University of Seville, 41009 Seville, Spain; 3Instituto de Biomedicina de Sevilla (IBIS), 41013 Seville, Spain

**Keywords:** osteoarthritis, phonophoresis, inflammation, ultrasound

## Abstract

Knee osteoarthritis (OA) is the most common joint disease. The administration of nonsteroidal anti-inflammatory drugs (NSAIDs) by phonophoresis is a therapeutic alternative to relieve pain in inflammatory pathologies. The main aim was to analyze the efficacy of the application of NSAIDs by phonophoresis in knee OA. A systematic review and meta-analysis of controlled clinical trials were performed between January and March 2021 in the following databases: Web of Science, Scopus, PubMed, Cinahl, SciELO, and PEDro. The PEDro scale was used to evaluate the level of evidence of the selected studies. The RevMan 5.4 statistical software was used to obtain the meta-analysis. Eight studies were included, of which five were included in the meta-analysis, involving 195 participants. The NSAIDs used through phonophoresis were ibuprofen, piroxicam, diclofenac sodium, diclofenac diethylammonium, ketoprofen, and methyl salicylate. The overall result for pain showed not-conclusive results, but a trend toward significance was found in favor of the phonophoresis group compared to the control group (standardized mean difference (SMD) = −0.92; 95% confidence interval: −1.87–0.02). Favorable results were obtained for physical function (SMD = −1.34; 95% CI: −2.00–0.68). Based on the selected studies, the application of NSAIDs by phonophoresis is effective in relieving the symptoms of knee OA. Future long-term studies are recommended.

## 1. Introduction

Knee osteoarthritis (OA) is an arthritic condition that causes pain, weakness, decreased function [1,2], and different gait impairments [3], overweight being one of the risk factors [4]. There are different physical treatments (extracorporeal shockwaves therapy, iontophoresis, physical activity, and hybrid hyaluronic acid injection) that have demonstrated their efficacy against this pathology [5,6,7,8]. The use of oral nonsteroidal anti-inflammatory drugs (NSAIDs) is often used widely and indiscriminately, sometimes as a symptomatic treatment, despite the gastrointestinal disturbances it can cause [9,10]. The topical use of this drug in the elderly avoids these undesirable effects [11]. An alternative to make topical application more efficient is through the use of ultrasound, a process called phonophoresis or sonophoresis [7].

Phonophoresis provides higher local concentrations of the drug than with simple topical application, increasing permeability through structural changes in the skin, as well as through the convection mechanisms inherent to the ultrasound effect [12]. The acoustic cavitation mechanism inherent to this type of application is the main cause of improved drug migration through the skin [13]. The combined use of the drug with skin penetration enhancers of substances called dendrimers [14] further increases permeability [15] and takes advantage of the analgesic and anti-inflammatory effect provided by the action of ultrasound on the organism [16,17]. This procedure has been widely used since 1954, without observing significant systemic adverse effects in the short and long term; only minor adverse effects on the skin have been reported in the scientific literature [18,19]. Several studies described the efficacy of the phonophoresis application with respect to different types of drugs administered [20,21,22,23] on diseases of different etiology [24,25,26,27].

Although there are reviews on the efficacy of ultrasound in knee OA [28], to the best of our knowledge, there is currently no evidence through meta-analysis analyzing the efficacy of phonophoresis with NSAIDs. Therefore, the main aim of this study was to analyze the efficacy of the application of NSAIDs by phonophoresis in knee OA.

## 2. Materials and Methods

### 2.1. Search Strategy

The search was performed following the Preferred Reporting Items for Systematic Revies and Meta-Analyses (PRISMA) [29] guidelines for systematic. The protocol of this systematic review and meta-analysis was registered in PROSPERO with the registration number CRD42021250328. The search was carried out between January and March 2021 in the following databases: Web of Science (WoS), PubMed, Scopus, Physiotherapy Evidence Database (PEDro), CINHAL, and SciELO, including all articles published since 2010. The following terms combined with Boolean operators were used: “Phonophoresis”, “Drugs”, “Ions”, “inflammation”, and “Pharmaceutical preparations”. In PubMed, Medical Subjects Headings (MeSH) descriptors were used. The search was restricted to controlled clinical trials. No language filters were used. The search strategy is shown in Appendix A.

### 2.2. Eligibility Criteria

The primary inclusion criteria were based on the PICO model [30]: (I) Population: subjects with knee OA; (II) Intervention: treatment with NSAIDs phonophoresis; (III) Comparison: placebo and other treatments; (IV) Outcomes: related to knee pain and functionality. The exclusion criteria were studies applying the simultaneous combination of phonophoresis with other treatments. Other inclusion criteria were randomized control trials, English and Spanish language, and trials only on humans. Exclusion criteria were quasi-experimental studies and animal testing research.

### 2.3. Study Selection and Data Extraction

First, the search was carried out in the different databases and duplicated articles were excluded. After that, titles and abstracts were screened to include those articles that did meet the eligibility criteria. Subsequently, the selected articles were finally included in the systematic review. Two reviewers (F.J.M.V. and G.G.M.) carried out the study selection process, review, and systematic data extraction. A third reviewer (D.L.A.) participated in achieving consensus in case of controversy.

The data extracted from each of the articles included in this review were number of participants, gender, age, mean body mass index (BMI), radiological assessment, type of intervention, duration, number of sessions, and frequency of the intervention, as well as measuring instruments and results.

### 2.4. Methodological Quality of the Studies

The PEDro scale [31] was applied in order to assess the methodological quality. This scale comprises 11 items related to the domains of selection, performance, detection, information, and attribution bases. Each item is scored with one point if the study meets the criteria. The first criterion was not used in the final calculation. A score greater than or equal to 6 is considered evidence level 1 (10–9: excellent; 8–6: good) and a score less than or equal to 5 is considered evidence level 2 (5–4: fair; less than 4: poor) [32].

### 2.5. Statistical Analysis

The Review Manager (RevMan) version 5.3. (the Cochrane Collaboration, the Nordic Cochrane Centre, København, Denmark) software was used to perform the meta-analysis. For the meta-analyses, the studies were separated into subgroups according to the measuring instrument used. The standardized mean difference (SMD) with a confidence interval (CI) of 95% and a value of *p* < 0.05 was used. Chi-square tests and the I2 value were used to determine heterogeneity. Due to the heterogeneity, the random effect model was used. Finally, the results were presented in forest plots.

The publication bias was calculated by the EPIDAT 4.2 software (Consellería de Sanidade, Xunta de Galicia, Spain; Organización Panamericana de la salud (OPS-OMS); Universidad CES, Colombia). The values of the Begg and Egger tests were used. A value of *p* < 0.01 was taken as a reference to consider that there was no publication bias. In addition, funnel plots were created.

## 3. Results

### 3.1. Study Included in the Systematic Review

A total of 328 potentially relevant articles were retrieved. In the screening period, 40 articles were excluded according to the inclusion/exclusion criteria; the exclusion causes are reported in the flowchart (Figure 1). From the eligibility section, 113 studies were excluded (not OA or NSAIDs topic). Finally, three studies were excluded in the qualitative synthesis because the measuring and protocols were off-topic from the present review. Eight articles analyzing the efficacy of NSAIDs phonophoresis for reducing the painful process in people with knee OA were included in the systematic review, and five in the meta-analysis (Figure 1).

A total of 195 participants were involved. The average sample size was 62 participants (range between 40 to 101). The age range of the participants was between 40 and 70 years, with a clear predominance of the female gender. Of the eight articles, five of them recorded a mean BMI of 30 kg/m^2^ or more, two articles between 24.1 and 26.3 kg/m^2^ on average, and in one article these data were absent. Regarding the diagnostic elements used for knee OA, seven studies used the criteria of the American Rheumatology Association [33] and one did not specify [34]. Five of the studies described the degree of radiological impairment of the participants according to the Kellgren and Lawrence (KL) classification [35], with a predominance of minimal and moderate grade. Three studies identified a mean duration of symptoms in the range of 4.4 to 7.6 months, two studies between 3.5 and 4.6 years, and two studies did not specify. The assessment systems used were predominantly the visual analog scale (VAS) for pain rating [36] and the Western Ontario and McMaster Universities Osteoarthritis Index (WOMAC) for pain and function [37,38], except for one study [39] that used only VAS. Along with the abovementioned assessment systems, some studies used other assessment systems such as range of mobility assessment [40], Lequesne functional index [41], Stanford health assessment questionnaire (HAQ) [42], fifteen minutes walking and a physician-patient interpersonal assessment [43] or twenty minutes timed brisk walking [44], and range of mobility with goniometry [34]. From the systematic review process, all the included studies reported the NSAIDs phonophoresis effectiveness for reducing the painful process in people with knee OA; nevertheless, not all the included studies showed benefits against short wave [43], ultrasound and TENS [45], and ultrasound alone [46]. The main characteristics of the studies included in the systematic review are shown in Table 1.

Regarding the methodological quality, three studies were rated excellent (one study: 10/10; two studies: 9/10) and five studies were rated good (one study: 8/10; two studies: 7/10; two studies: 6/10). Table 2 shows the methodological quality of the studies.

### 3.2. Study Groups Included in the Meta-Analysis

A total of five studies [34,40,43,46,47] were included in the meta-analysis according to two different instruments: VAS and WOMAC. Moreover, the VAS group was divided into two subgroups: (i) VAS for pain at rest, and (ii) VAS for pain during movement. The WOMAC group was also divided into four groups: three subgroups according to their different subscales: (i) pain, (ii) physical functioning, and (iii) stiffness; and the last group analyzing the total score. A high degree of heterogeneity (I2 > 50%) for both outcomes (VAS and WOMAC) was found.

Four studies [40,43,46,47] were included in the VAS for pain at rest subgroup. All the studies showed improvements. The study by Jun et al. [40] obtained the best results. The overall result of the meta-analysis was not conclusive, but a trend toward significance was found for the phonophoresis group compared to the control group (std. mean difference = −0.92 [−1.87, 0.02]; confidence interval 95% (see Figure 2). Two studies [40,47] were included in the VAS for the pain during movement subgroup, obtaining dissenting results. The overall result of the meta-analysis was not conclusive.

Finally, the overall result for the VAS group showed not-conclusive results, but a trend toward significance: SMD = −0.92; 95% CI: −1.87 to 0.02 was found in favor of the experimental group compared to the control group (placebo or alternative treatment). The forest plots are shown in Figure 2.

All the studies were included in the WOMAC for pain subgroup, showing improvements, except for the study by Oktayoǧlu et al. [47]. The study by Jun et al. [40] obtained the best results. The overall result of the meta-analysis was favorable for the phonophoresis group compared to the control group. Four studies [34,40,43,47] were included in the WOMAC for physical function subgroup, obtaining similar results to the previous subgroup. The overall result of this meta-analysis was not conclusive, but a trend toward significance was found for the phonophoresis group. In addition, the previous studies [34,40,43,46,47] were included in the WOMAC for stiffness subgroup. The results obtained were diverse and the overall result of this meta-analysis was not conclusive. In addition, four studies [40,43,46,47] measured the effects through the WOMAC total score. All the studies showed improvements, except for the study by Oktayoǧlu et al. [47]. The overall result of this meta-analysis was not conclusive, but a trend toward significance was found for the phonophoresis group compared to the control group.

Finally, the overall result for the WOMAC group showed favorable results for the phonophoresis group compared to the control group: SMD = −1.34; 95% CI: −2.00 to −0.68. The forest plots are shown in Figure 3.

Regarding the risk of bias, the results of the Begg and Egger tests (Table 3) and funnel plots showed no significant publication bias (Figure 4 and Figure 5).

## 4. Discussion

All the studies included in the review analyzed the efficacy of the NSAIDs phonophoresis application in knee OA. The parameters preferably used in the application of phonophoresis were the continuous modality, 1 MHz frequency, intensity of 1 or 1.5 W/cm^2^, and an application time ranging from 5 to 10 min; except Jun et al. [40], which used a frequency of 40 KHz applied for 30 min. The nonuse of high sonophoretic intensities and times prevents the occurrence of adverse reactions on the application area [48], as well as the use of 1 MHz frequency, and increases the depth of action of the ultrasonic wave [49]. However, the use of continuous modality may be controversial. Continuous modality is effective and recommended in chronic pathologies such as knee OA [17] and its thermal effect increases the permeability of the skin, as long as it does not cause tissue damage or damage to the structure of the drug administered [50]. In addition, in the pulsing mode, the power is reduced, taking into account that the higher the power, the greater the depth [49]. However, the study conducted by Aldwaikat et al. [51] showed that an increase in amplitude in continuous mode causes detachment of the stratum corneum; a situation that does not occur when using pulsed mode. Yin et al. [52] described the same efficacy of the continuous and pulsed mode when using dual frequency, avoiding damage on the corneal surface by excessive thermal stimulation. Although most studies used the aforementioned parameter, it is not observed to have a direct relationship with the results obtained when comparing its efficacy with other therapeutic alternatives. Luksurapan et al. [46] and Boonhong et al. [45] observed greater efficacy of phonophoresis with piroxicam versus ultrasound alone or combined with TENS; in contrast, Monisha et al. [39] did not find significant differences, although all three studies used the same parameters for phonophoresis.

When comparing the administration of NSAIDs through phonophoresis with other therapeutic alternatives, only four studies [34,39,44,47] found statistically significant differences in favor of its use as the most effective technique to reduce pain. Although these studies describe a level of significance similar to the rest of the studies (*p* < 0.001 and *p* < 0.05), the methodological quality observed was greater in those that did not find statistically significant differences. Regarding the stiffness, two of the included studies [34,40] found no significant differences in efficacy when compared to other forms of treatment.

The review conducted by Wang et al. [5], which was focused on shock wave therapy, found a greater efficacy on pain, stiffness, and function evaluated with WOMAC, and pain evaluated with VAS, compared to the results obtained in our meta-analysis. One of the reasons could be related to the longer time of the treatment sessions and follow-up (two sessions per week during five weeks of treatment and six months of follow-up). Therefore, it could be suggested that the time of treatment may be an important factor with respect to decreased pain and increased function. Similarly, increasing the time of application in each session could be another factor to consider, as shown by the best results obtained by Jun et al. [40]. Another alternative to pain and lack of function is therapeutic exercise, except that its effectiveness is mainly in young people who do not require surgical intervention [53].

Although the number of participants included in the selected studies does not represent a representative sample of the population incidence of this type of injury [54], the data extracted with respect to age, gender, and BMI are in line with the studies carried out by Farreras and Rozman [55]. Regarding gender, the average percentage of subjects is approximately 84% of individuals belonging to the female gender, compared to 15% of the male gender. This percentage difference between genders could be due to a lower adherence to treatment by men compared to women [46,56], an issue that would prevent the extrapolation of the results obtained to this population gender [57]. Regarding the BMI, six of the eight studies agree with Farreras and Rozman [55] in the overweight factor as a possible variable to consider in this type of pathology. These studies describe the overweight of their participants as grade I and II according to the classification established by the World Health Organization (WHO) [58]. Only the study by Boonhong et al. [45] described a mean weight of 24.1 kg/m^2^ BMI in the subjects studied.

The common use in most of the studies included in this review of the VAS and WOMAC assessment systems, diagnostic elements (American Rheumatology Association Criteria), and analysis of radiological deterioration (KL) would allow this work to be used as a comparative reference for future studies, except for the limitations described by the authors of these studies.

Most of the studies performed measurements before–after applying the treatment. Only two studies followed up the study subjects. Jun et al. [40] repeated the measurement four weeks after the end of treatment, finding no significant differences with other therapeutic alternatives. In contrast, Oktayoǧlu et al. [47] repeated the measurement at one month, two months, and three months post-treatment, obtaining significant differences during gait. It should be noted that the patients included in the study by Oktayoǧlu et al. [47] presented high level of radiological deterioration (KL) and a longer duration of symptoms.

From the statistically significant differences in favor of phonophoresis versus conventional ultrasound therapy, some studies [34,39,47] showed the importance of the substance administered versus ultrasound as a tool to facilitate its absorption. The NSAIDs used in the studies were ibuprofen, piroxicam, diclofenac sodium, diclofenac diethyl-ammonium, ketoprofen, and methyl salicylate. The more repeated use of diclofenac and piroxicam may be due to their efficacy [59,60]. However, Dogruyol et al. [61] showed greater efficacy of ibuprofen gel versus piroxicam gel in elderly subjects. All active ingredients were used in different proportions mixed with a conductive substance in gel form for administration. The preference of the gel form over cream may be due to the better transmissibility of the former in application by phonophoresis [44,62,63]. Three of the included studies used other therapeutic techniques in addition to the techniques under study. The use of ultrasound combined with TENS [45] could increase the analgesic effect of the combination [64]. Similarly, the prior application of heat [34,43] to facilitate transdermal administration of the drug [19,65], could influence the comparison of results. To avoid such bias, Akinbo et al. [34] justifies the evaluation performed two days after the end of the treatment in order to compensate for the thermal effect caused by the previous application of heat. 

Some limitations need to be addressed. The results should be taken with caution due to the limited number of studies included in the meta-analysis. It was not possible to statistically compare all the studies included in the systematic review because different measuring instruments were used. Furthermore, the protocols used by the studies were heterogeneous, including different time measurement and follow-up periods, substances administered, and phonophoresis parameters, making comparisons difficult. Therefore, it is necessary to carry out randomized controlled trials unifying protocols regarding the substance administered, phonophoresis parameters, intervention duration, and measuring instruments. This could provide more information on which specific aspects of the intervention might have the greatest impact on outcomes. As a strength, to the best of our knowledge, this is the first scientific document to study the NSAIDs phonophoresis effectiveness in OA, with qualitative and quantitative analysis, only focused on randomized control trails.

## 5. Conclusions

In view of the results obtained, we conclude that the application of phonophoresis with NSAIDs is effective as a treatment for knee OA, although it does not appear to be more effective than other therapeutic alternatives.

Future studies are recommended with applications of longer time per session, treatment time, and follow-up in order to observe if the increase of these factors increases their efficacy. Other outcomes to consider in future studies will be the substance administered, phonophoresis parameters, intervention duration, and measuring instruments.

## Figures and Tables

**Figure 1 biomedicines-10-03254-f001:**
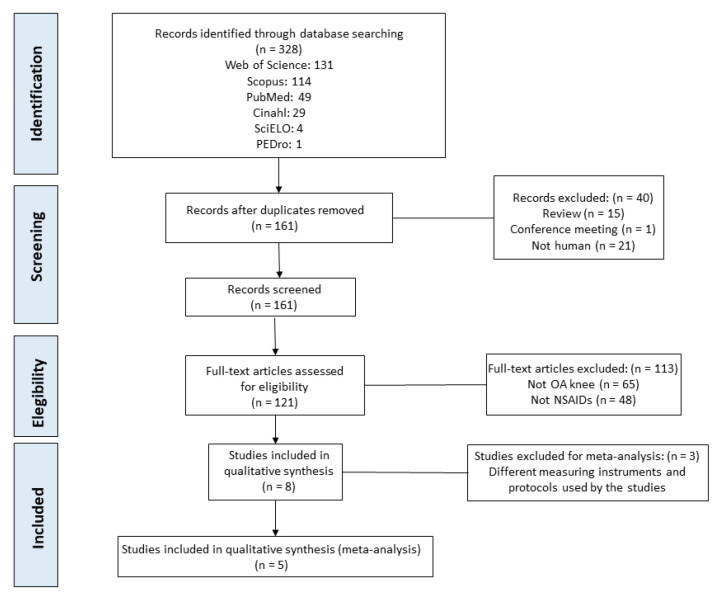
Flow diagram of the systematic review and meta-analysis.

**Figure 2 biomedicines-10-03254-f002:**
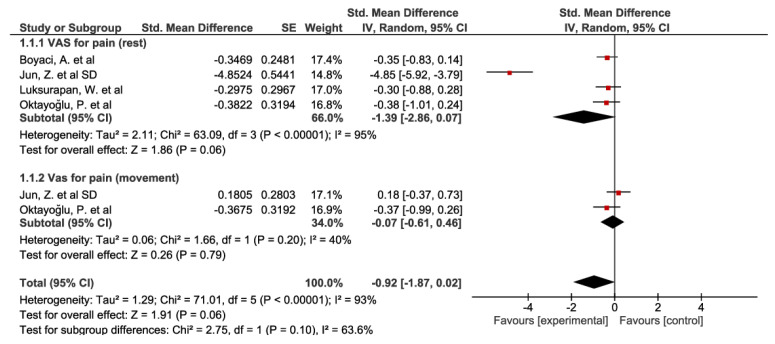
Forest plot for visual analogue scale. The red squares indicate the weight assigned to the study. The horizontal lines depict the confidence interval. The black rhombuses show the overall result.

**Figure 3 biomedicines-10-03254-f003:**
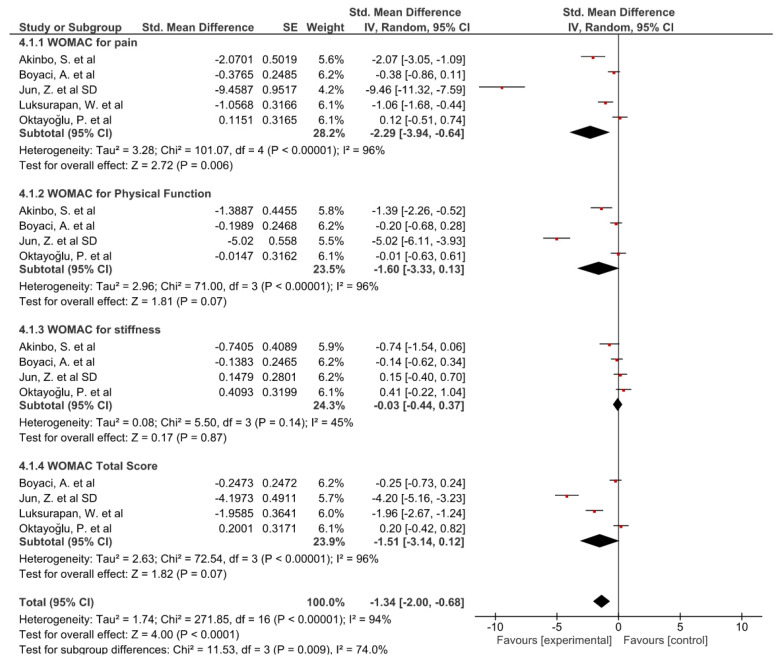
Forest plot for Western Ontario and McMaster Universities Osteoarthritis Index. The red squares indicate the weight assigned to the study. The horizontal lines depict the confidence interval. The black rhombuses show the overall result.

**Figure 4 biomedicines-10-03254-f004:**
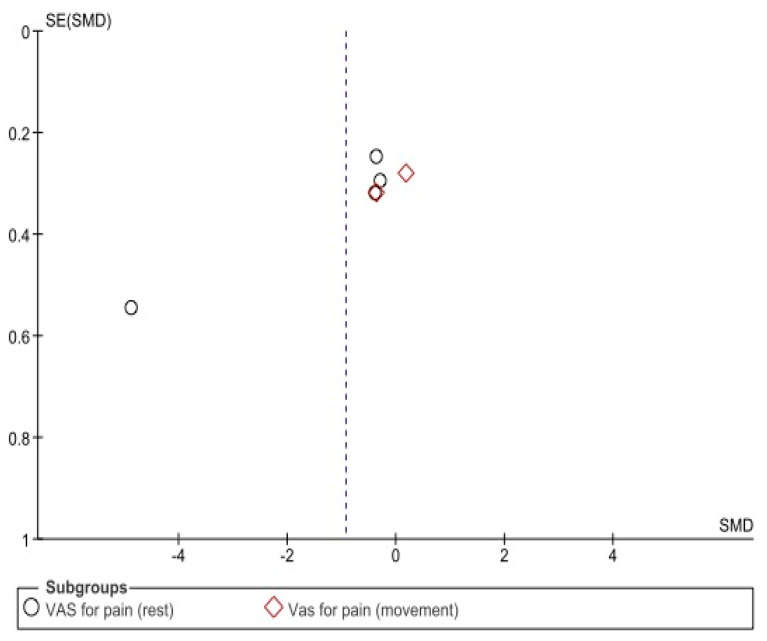
Funnel plot showing publication bias for visual analogue scale.

**Figure 5 biomedicines-10-03254-f005:**
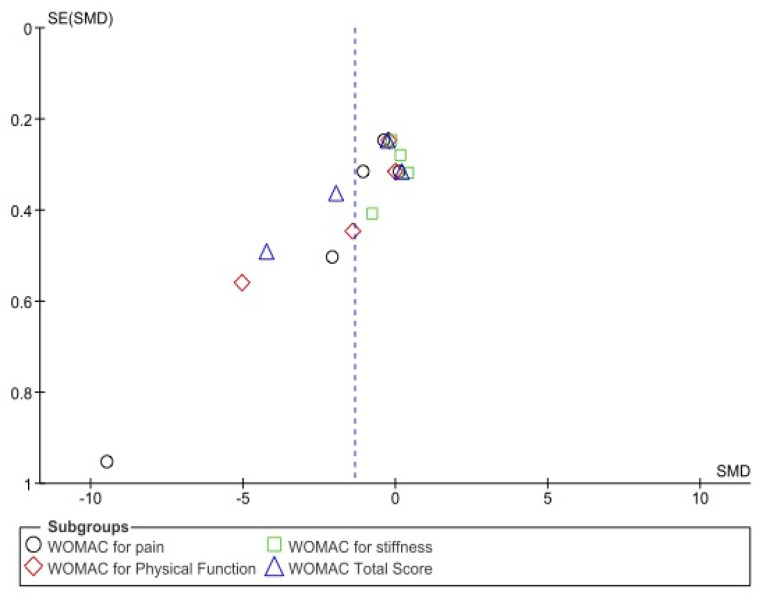
Funnel plot showing publication bias for Western Ontario and McMaster Universities Osteoarthritis Index.

**Table 1 biomedicines-10-03254-t001:** Main characteristics of the studies included in the systematic review.

Author	Sample	Intervention	Ultrasonic Parameters	Measuring Instruments	Results
Belindayi et al. [44] (2018). Randomized, single-blind, comparative study.	n = 61 (54 women and 7 men) Average age: 57.9 years Average BMI: 30.1 kg/m^2^	Group Php +Ibuprofen 5% (gel) (n = 30). Group Php +Ibuprofen 5% (cream) (n = 31). 5 days/2 weeks. Applied to the most symptomatic knee.	Continuous mode, frequency: 1 MHz, intensity: 1 W/cm^2^, and application time: 5 min.	VAS WOMAC	Both groups improve significantly, with greater improvement in the Php + Ibuprofen (gel) group (*p* < 0.001).
Boonhong et al. [45] (2018). Comparative, randomized, controlled, double-blind trial.	N = 61 (55 women and 6 men) Average age: 63.4 years Average BMI: 24.2 kg/m^2^ Average duration of symptoms: 4.4 to 7.6 months KL: Grade I-II-III	Group Php + Pyroxicam 0.5% (gel) + simulated TENS (n = 30). Group Ultrasound + nonsimulated TENS (n = 31). 5 days/2 weeks. Applied to the most symptomatic knee.	Continuous mode, frequency: 1 MHz, intensity: 1 W/cm^2^, and application time: 10 min.	VAS WOMAC	Both interventions were effective. Observing greater improvement in the Php group with Piroxicam, although without statistically significant differences (*p* < 0.001).
Monisha et al. [39] (2018). Double-blind, randomized, controlled trial.	N = 50 women Age range: 40–70 years KL: Grade II-III	Group Php + Piroxicam (gel) + coupling gel (ratio 4:10). Php + Dimethyl Sulfoxide Group. Ultrasonic Group. 5 days/2 weeks. Applied on both knees.	Continuous mode, frequency: 1 MHz, intensity: 1 W/cm^2^, and application time: 10 min.	VAS	Treatment with Php + Piroxicam (gel) provides better pain relief than ultrasound in mild to moderate osteoarthritis of the knee (*p* < 0.00).
Jun et al. [40] (2015). Randomized, double-blind, placebo-controlled trial.	N = 96 (77 women and 19 men) Average age: 60 years Average BMI: 30.8 kg/m^2^ Average duration of symptoms: 7.2 months	Group Php + Chinese medicine substance (n = 38). Group Php +Diclofenac sodium (gel) 10 mg (n = 39). Placebo group (n = 19).	Low-frequency Php (40 kHz), application time: 30 min/session.	VAS WOMAC ROM	Significant improvement in pain and physical function in the Php treatment groups versus the placebo group. No significant differences were observed between the two groups with respect to the improvement of stiffness and range of motion (*p* < 0.05).
Oktayoglu et al. [47] (2014). Randomized controlled study.	N= 40 (30 women and 10 men) Average age: 54.8 years Average BMI: 29.9 kg/m^2^ Mean duration of symptoms: 4.6 years KL: Grade II-III-IV	Group Php + Diclofenac diethylammonium 1.16% (gel). Ultrasonic Group. 5 days/2 weeks. Applied on both knees.	Continuous mode, frequency: 1 MHz, intensity: 1.5 W/cm^2^, and application time: 10 min.	VAS WOMAC Lequesne Functional Index HAQ	Both treatments are effective. Greater statistically significant improvement in pain on walking in the Php + Diclofenac diethylammonium (gel) group (*p* < 0.05).
Boyaci et al. [43] (2013). Randomized, single-blind, comparative study.	N= 101 women Average age: 51.9 years Mean BMI: 32.8 kg/m^2^ Average duration of symptoms: 3.5 years KL: Grade II-III	Group Php + Ketoprofen (gel) 100 mg (n = 33). Ultrasound Group (n = 33). Short wave group (n = 35). Previously hot compresses 20 min in all the groups. 5 days/2 weeks. Applied on both knees.	Frequency: 1 MHz, intensity: 1.5 W/cm^2^, and application time: 8 min.	VAS WOMAC Gait Physician-patient assessment	The three treatment modalities were effective, with no significant differences between them (*p* < 0.001).
Luksurapan et al. [46] (2013). Randomized, controlled, double-blind trial.	N= 46 (45 women and 1 man) Average age: 58.9 years Mean BMI: 26.3 kg/m^2^ Mean duration of symptoms: 3.3 years KL: Grade I-II-III	Group Php + Pyroxicam 0.5% (gel) (n = 23). Ultrasound Group (n = 23). 5 days/2 weeks. Applied to the most symptomatic knee.	Continuous mode, frequency: 1 MHz, intensity: 1 W/cm^2^, and application time: 10 min.	VAS WOMAC	Greater efficacy for pain and function in the Php +Piroxicam group. No statistically significant differences between the two groups (*p* < 0.001).
Akinbo et al. [34] (2011). Randomized controlled study.	N= 40 (34 women and 6 men) Average age: 57.5 years Average BMI: 31 kg/m^2^	Php Group + Diclofenac sodium 1% (gel) (n = 14). Group Php + Methyl Salicylate 15% (gel) (n = 14). Ultrasound group (n = 12). Previous application of heat (15 min) and ergometer for all groups. 5 days/2 weeks. Applied on painful knee or right knee if bilateral.	Continuous mode, frequency: 1 MHz, intensity: 1 W/cm^2^, and application time: 5 min.	WOMAC Gait ROM	All groups obtained improvement except in stiffness. The Php + Diclofenac Sodium group had a statistically significant improvement over the other two groups (*p* < 0.05).

BMI, body mass index; HAQ, Health Assessment Questionnaire; KL, Kellgren and Lawrence; Php, phonophoresis; ROM, range of motion; VAS, visual analogue scale; WOMAC, Western Ontario and McMaster Universities Osteoarthritis Index.

**Table 2 biomedicines-10-03254-t002:** The methodological quality of the studies (PEDro scale).

Author	C1	C2	C3	C4	C5	C6	C7	C8	C9	C10	C11	Total
Belindayi et al. [44] (2018)	1	1	1	1	0	0	1	1	1	1	1	8/10
Boonhong et al. [45] (2018)	1	1	1	1	1	0	1	1	1	1	1	9/10
Monisha et al. [39] (2018)	1	1	1	1	1	1	0	0	0	1	0	6/10
Jun et al. [40] (2015)	1	1	1	1	1	1	1	1	0	1	1	9/10
Oktayoglu et al. [47] (2014)	1	1	1	1	0	0	0	1	1	1	1	7/10
Boyaci et al. [43] (2013)	1	1	1	1	0	0	1	1	0	1	1	7/10
Luksurapan et al. [46] (2013)	1	1	1	1	1	1	1	1	1	1	1	10/10
Akinbo et al. [34] (2011)	1	1	1	1	0	0	0	1	0	1	1	6/10

C1: Choice criteria were specified. C2: Subjects were randomly assigned to groups (in a crossover study, subjects were randomly distributed as they received treatments). C3: Allocation was concealed. C4: Groups were similar at baseline with respect to the most important prognostic indicators. C5: All subjects were blinded. C6: All therapists administering therapy were blinded. C7: All evaluators who measured at least one key outcome were blinded. C8: Measures of at least one of the key outcomes were obtained from more than 85% of the subjects initially assigned to the groups. C9: Results were presented for all subjects who received treatment or were assigned to the control group, or when this could not be achieved, data for at least one key outcome were analyzed on an intention-to-treat basis. C10: Results of statistical comparisons between groups were reported for at least one key outcome. C11: The study provides point-in-time and variability measures for at least one key outcome.

**Table 3 biomedicines-10-03254-t003:** Results of Begg and Egger tests.

Title 1	Begg	Egger
VAS REST	0.3082	0.0018
VAS MOVEMENT	1.0000	-
WOMAC (PAIN)	0.2207	0.0439
WOMAC (STIFFNESS)	0.7341	0.8171
WOMAC (PHYSICAL FUNCTION)	0.3082	0.0700
WOMAC TOTAL	0.3082	0.1599

VAS, visual analogue scale; WOMAC, Western Ontario and McMaster Universities Osteoarthritis Index.

## Data Availability

Not applicable.

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
