# Peer review of "Phonophoresis through Nonsteroidal Anti-Inflammatory Drugs for Knee Osteoarthritis Treatment: Systematic Review and Meta-Analysis"

_biomedicines, 2022, doi:10.3390/biomedicines10123254_

Round 1

Reviewer 1 Report

I thank the editors for giving me the opportunity to review this manuscript. I carefully read this article which aims to evaluate the efficacy of the application of NSAIDs by phonophoresis in knee osteoarthritis, through a meta-analysis and a systematic review. I think some corrections are needed, which I list below: ABSTRACT - Line 14: “Knee osteoarthritis (OA) is the most common joint disease”. The sentence is not relevant. Insert it at the beginning of the abstract. INTRODUCTION - Line 33: “There are different physical treatments that have demonstrated their efficacy against this pathology [5–7]”. What other physical therapies are you referring to? Mention them briefly. - Line 48: “Several studies described the efficacy of the phonophoresis application with respect to different types of drugs administered [19–22] on diseases of different etiology [23–26]”. Since the article deals with the use of phonophoresis in knee arthrosis, you should introduce the problem of your research by quoting some studies that evaluated phonophoresis in osteoarthritis and then in particular in knee arthrosis. METHODS - Broaden the inclusion and exclusion criteria. For example, were articles written in any language considered? RESULTS - Line 106: “Eight articles analyzing the efficacy of NSAIDs phonophoresis for reducing the painful process in people with knee OA were included in the systematic review and five in the meta-analysis”. You should describe in the text (as well as in the figure) why of the 328 articles only 8 were selected for systematic review and 5 for meta-analysis. - Only the results of the meta-analysis are reported but no mention is made of the results of the systematic review. Was only one meta-analysis performed?? Motivate - Line 162: “The overall result of the meta-analysis was not conclusive, but a trend toward significance was found for the phonophoresis group compared to the control group”. Data such as standardized mean difference (SMD) and confidence interval should be added. Add this data also in the rest of the text. DISCUSSION - To be reviewed. Why do you compare this meta-analysis on phonorhesis in knee osteoarthritis with a meta-analysis on shock waves? - Line 266: I would insert the entire paragraph at the beginning of the discussion. - Does your studio have any strengths? If present, add them. Add the following references in the text: PMC8619194; https://doi.org/10.3390/app11188711;

Author Response

Itemized List of Changes from the Reviewers and Editorial

Dear Editor

Please find a revision of our manuscript entitled “Phonophoresis through Nonsteroidal anti-inflammatory drugs for knee osteoarthritis treatment: systematic review and meta-analysis(Biomedicines-2086478). We would like to thank the Reviewers for their thoughtful and constructive comments. We have considered all suggestions and have incorporated them into the revised manuscript. A manuscript has been uploaded with tracked-changes. We believe our manuscript is stronger as a result of the modifications. An itemized point-by-point response to the Reviewers’ comments is presented below.

Reviewer 1:

Reviewer comments (RC): I thank the editors for giving me the opportunity to review this manuscript. I carefully read this article which aims to evaluate the efficacy of the application of NSAIDs by phonophoresis in knee osteoarthritis, through a meta-analysis and a systematic review. I think some corrections are needed, which I list below:

RC: ABSTRACT - Line 14: “Knee osteoarthritis (OA) is the most common joint disease”. The sentence is not relevant. Insert it at the beginning of the abstract.

Authors Answer (AA): We would like to thank the reviewer for the thoughtful and constructive comments and contribution. The Abstract section has been modified according to the reviewer comment in the describes lines.

RC: INTRODUCTION - Line 33: “There are different physical treatments that have demonstrated their efficacy against this pathology [5–7]”. What other physical therapies are you referring to? Mention them briefly.

AA: Thank you very much for your suggestion at this point. We have modified the statement including these recommendations.

RC: Line 48: “Several studies described the efficacy of the phonophoresis application with respect to different types of drugs administered [19–22] on diseases of different etiology [23–26]”. Since the article deals with the use of phonophoresis in knee arthrosis, you should introduce the problem of your research by quoting some studies that evaluated phonophoresis in osteoarthritis and then in particular in knee arthrosis.

AA: Thank you very much for the suggestion. We agreed with the reviewer. We have revised in deep all the references. In this sense, the following references were related with the knee OA:

  1. Heyadati, R.; Aminian-Far, A.; Darbani, M.; Al., E. Efficacy of Glucosamine Compounds Phonophoresis in Knee Osteoarthritis. Koomesh 2016, 18, 276–285.
  2. Toopchizadeg, V.; Javadi, R.; Sadat, B.E. Therapeutic Efficacy of Dexamethasone Phonophoresis on Symptomatic Knee Osteoarthritis in Elderly Women. International Journal of Women’s Health and Reproduction Sciences 2014, 2, 168–177, doi:10.15296/ijwhr.2014.25.

The remaining included references have been included in order to contextualize the information in the introduction section.

RC: METHODS - Broaden the inclusion and exclusion criteria. For example, were articles written in any language considered?

AA: We would like to thank the reviewer for the thoughtful and constructive comments and contribution. The “Elegibility criteria” section has been modified according to the reviewer suggestion.

RC: RESULTS - Line 106: “Eight articles analyzing the efficacy of NSAIDs phonophoresis for reducing the painful process in people with knee OA were included in the systematic review and five in the meta-analysis”. You should describe in the text (as well as in the figure) why of the 328 articles only 8 were selected for systematic review and 5 for meta-analysis.

AA: Thank you very much at this point. The figure 1 and the main text has been modified according to the reviewer suggestion. A new paragraph was included in the 3.1. subsection. 

RC: Only the results of the meta-analysis are reported but no mention is made of the results of the systematic review. Was only one meta-analysis performed?? Motivate

AA: Thank you very much for the suggestion. The systematic review result has been shown and reported in the 3.1 subsection, between the 115 and 173 lines, and included the main text, the figure 1 and tables 1 and 2. In the other hand the meta-analysis results has been reported in the new subsection 3.2.

RC: Line 162: “The overall result of the meta-analysis was not conclusive, but a trend toward significance was found for the phonophoresis group compared to the control group”. Data such as standardized mean difference (SMD) and confidence interval should be added. Add this data also in the rest of the text.

AA: Thank you very much for the suggestion. The standard mean difference was included in the main document (Subsection: 3.2).

RC: DISCUSSION - To be reviewed. Why do you compare this meta-analysis on phonorhesis in knee osteoarthritis with a meta-analysis on shock waves?

AA: Thank you very much at this point. To the best of our knowledge, the present review is the first one on the NSAIDs phonophoresis effectiveness in OA. It has not been possible to find another review to compare the results in the same topic, pathology and administered drug. For this reason, we tried to compared with similar studies, closed to our topic review.

RC: Line 266: I would insert the entire paragraph at the beginning of the discussion.

AA: According to the reviewer’s suggestion, the paragraph was moved to the beginning of the Discussion section.

RC: Does your studio have any strengths? If present, add them.

AA: We would like to thank the reviewer for the thoughtful and constructive comments and contribution. A statement was added at the end of the Discussion section according to the present suggestion.

RC: Add the following references in the text: PMC8619194; https://doi.org/10.3390/app11188711;

AA: Thank you very much for your suggestion. The reference has been included at the beginning of the Introduction section (reference number 8).

Reviewer 2 Report

This paper mainly reviewed the efficacy of phonophoresis with NSAIDs for the knee OA treatment. Overall, the paper is well organized and designed.  Following is some suggestion.

(1)   Since the knee OA is a complex disease, which affects the whole joint and involves with the cartilage, subchondral bone, osteophyte, meniscus, and synovial fluid. It can cause joint pain and disability. As for the NSAIDS drugs could relieve pain or mitigate inflammation. However, in the paper, the author only focused on the pain relief in the publication, did the author find any effect for the inflammation during the treatment?

(2)   What is the risk of the therapy strategy of NSAIDs with phonophoresis for knee OA? Could the author clarify it in the paper?

(3)   What is the rationale for the Future long-term studies recommended? A lot of factors, including substance administered, phonophoresis parameters, intervention duration, and measuring instruments, can affect the treatment results. Why did the author point out the longer time rather than other factors? Could the author clarify it in more details? 

Author Response

Itemized List of Changes from the Reviewers and Editorial

Dear Editor

Please find a revision of our manuscript entitled “Phonophoresis through Nonsteroidal anti-inflammatory drugs for knee osteoarthritis treatment: systematic review and meta-analysis(Biomedicines-2086478). We would like to thank the Reviewers for their thoughtful and constructive comments. We have considered all suggestions and have incorporated them into the revised manuscript. A manuscript has been uploaded with tracked-changes. We believe our manuscript is stronger as a result of the modifications. An itemized point-by-point response to the Reviewers’ comments is presented below.

Reviewer 2:

This paper mainly reviewed the efficacy of phonophoresis with NSAIDs for the knee OA treatment. Overall, the paper is well organized and designed.  Following is some suggestion.

Reviewer comments (RC): Since the knee OA is a complex disease, which affects the whole joint and involves with the cartilage, subchondral bone, osteophyte, meniscus, and synovial fluid. It can cause joint pain and disability. As for the NSAIDS drugs could relieve pain or mitigate inflammation. However, in the paper, the author only focused on the pain relief in the publication, did the author find any effect for the inflammation during the treatment?

Authors Answer (AA): We would like to thank the reviewer for the thoughtful and constructive comments and contribution. According to the included studies in this review, none of these studies reported outcomes related with the inflammation during the treatment.

RC: What is the risk of the therapy strategy of NSAIDs with phonophoresis for knee OA? Could the author clarify it in the paper

AA: Thank you very much at this point. The introduction section (line 59-60) has been modified to clarify the information about adverse effects. Now it is possible to read information about significant and minor adverse effect in the main document.

RC: What is the rationale for the Future long-term studies recommended? A lot of factors, including substance administered, phonophoresis parameters, intervention duration, and measuring instruments, can affect the treatment results. Why did the author point out the longer time rather than other factors? Could the author clarify it in more details?

AA: Thank you very much, we are in full agreement with the suggestion. Accordingly, we added a statement at the end of the Conclusions section.